# Outreach and Impact of the Hyper-Kamiokande experiment in Mexico and Latin America from 2020 to 2023 in social media.

Judith Torres–Jiménez[1*], Montserrat Montiel–Dücker [1**], Eduardo de La Fuente [2†] and A. K. Tomatani-Sánchez[2††]

**1** Tecnológico de Monterrey, Escuela de Ingeniería y Ciencias, Av. Eugenio Garza Sada No. 2501 Sur, Tecnológico, 64700 Monterrey, N.L.
**2** Departamento de Física, CUCEI, Universidad de Guadalajara
**3** Tecnológico de Monterrey, Escuela de Ingeniería y Ciencias, Av. Gral Ramón Corona No. 2514, Colonia Nuevo México, 45201 Zapopan, Jal

* A00830020@tec.mx ,  ** A01642845@tec.mx, † eduardo.delafuente@academicos.udg.mx, †† katsumi@tec.mx

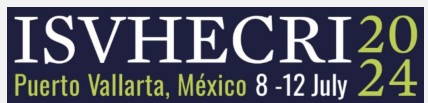

*22nd International Symposium on Very High Energy Cosmic Ray Interactions (ISVHECRI 2024) Puerto Vallarta, Mexico, 8-12 July 2024*

## Abstract

**Outreach Hyper-Kamiokande Mexico has significantly boosted the visibility of the project and increased Latin American engagement. This paper evaluates the strategies and impacts of its outreach initiatives, emphasizing digital communication. Data indicates a significant uptick in engagement metrics, signaling growing scientific interest, driven by an inclusive communication approach. These results highlight the importance of localized outreach within global collaborations, showing how targeted engagement can elevate public understanding and support for large-scale scientific projects in Latin America.**

## 1 Introduction

Mexico's official participation in Hyper-Kamiokande was established in 2019 through the collaborative efforts of PhD professors Saúl Cuen-Rochin (Instituto Tecnológico de Estudios Superiores de Monterrey; ITESM) and PhD. Eduardo de La Fuente (Universidad de Guadalajara; UdeG), with the support of PhD. Akira Konaka (TRIUMF Laboratory, Vancouver) and PhD. Takaaki Kajita (Institute of Cosmic Ray Research, University of Tokyo). Subsequently, in 2024 the Universidad Autónoma de Sinaloa (UAS) and the Universidad Autónoma de Chiapas (UNACH) joined the Hyper-K México collaboration (see also 1).

Outreach Hyper-Kamiokande Mexico (Hyper-K MX) is the Mexican branch of the Hyper-K neutrino experiment to be installed in Japan. The purpose is to connect the breakthroughs of

the Hyper-K project with Mexico and Latin America, fostering a deeper understanding of neutrino physics and its broader implications. Through social media engagement, collaborations with laboratories, public talks, and cultural and educational initiatives. Outreach Hyper-K MX has reached a diverse audience, promoted scientific literacy, and ignited interest in fundamental research throughout the region. This article examines the impact of our outreach efforts, highlighting our role in communicating advances related to this neutrino detector and the contributions made by UdeG and the ITESM (1) from 2020 to 2023, even though the dissemination of the project on social media began in 2021.

Monitoring social media engagement allows companies to gain insights into potential customers. In business marketing the return-on-investment (ROI) metric is essential to justify the money invested in marketing and the return on sales. While ROI is not an essential metric in scientific outreach, others remain indispensable (2). Key performance indicators (KPIs) such as audience reach and engagement rates allow us to evaluate the success of our outreach initiatives. Although a standardized metric across scientific accounts is not feasible, since every account addresses unique projects and objectives, customized metrics may provide a framework to understand engagement with science.

## 2  Methodolody

### 2.1  Research Approach

This research focuses on evaluating the outreach initiatives carried out on social media platforms by Outreach Hyper-K MX and comparing them with other outreach accounts on a national, institutional, and global scale. The approach adopted is quantitative, utilizing data collected from 2020 to 2023.

### 2.2  Data Collection and Processing

Due to API restrictions, we manually collected public data from YouTube, Facebook, X, Instagram, and TikTok. We processed the collected data using data analysis tools in Python. We generated graphs and visualizations to compare the metrics of Outreach Hyper-K MX with other global and national initiatives, as well as the general outreach efforts of Hyper-K.

### 2.3  Comparison of Initiatives

To conduct the comparison, we established three evaluation criteria based on the engagement on social media for each outreach initiative (2). Shares and comments are key metrics, as they indicate the value of the content(3).

The first metric assesses interaction growth by comparing total and normalized social media interactions across outreach efforts. Normalization is done by dividing yearly interactions by the number of institutions managing the outreach of each experiment.

The second metric is the Exploratory Data Analysis (EDA) (4). We used measures such as mean and standard deviation, to evaluate outreach performance describing social media data.

The third metric is the engagement of followers on each outreach initiative's social media. The engagement rate (5) is defined as:

$$\text{Engagement Rate} = \frac{\text{Average number of interactions}}{\text{Number of followers}} \times 100. \tag{1}$$

where interactions include comments, shares, likes, and views, among others.

## 2.4 Limitations

It is important to note that data collection through manual entry may present limitations, such as potential entry errors and changes in data access policies from social media platforms.

# 3 Social Media Impact Results

Figure 1: Anual interactions between HAWC (blue) Hyper-K MX (green), Hyper-K JP (red) and IceCube Observatory (black).

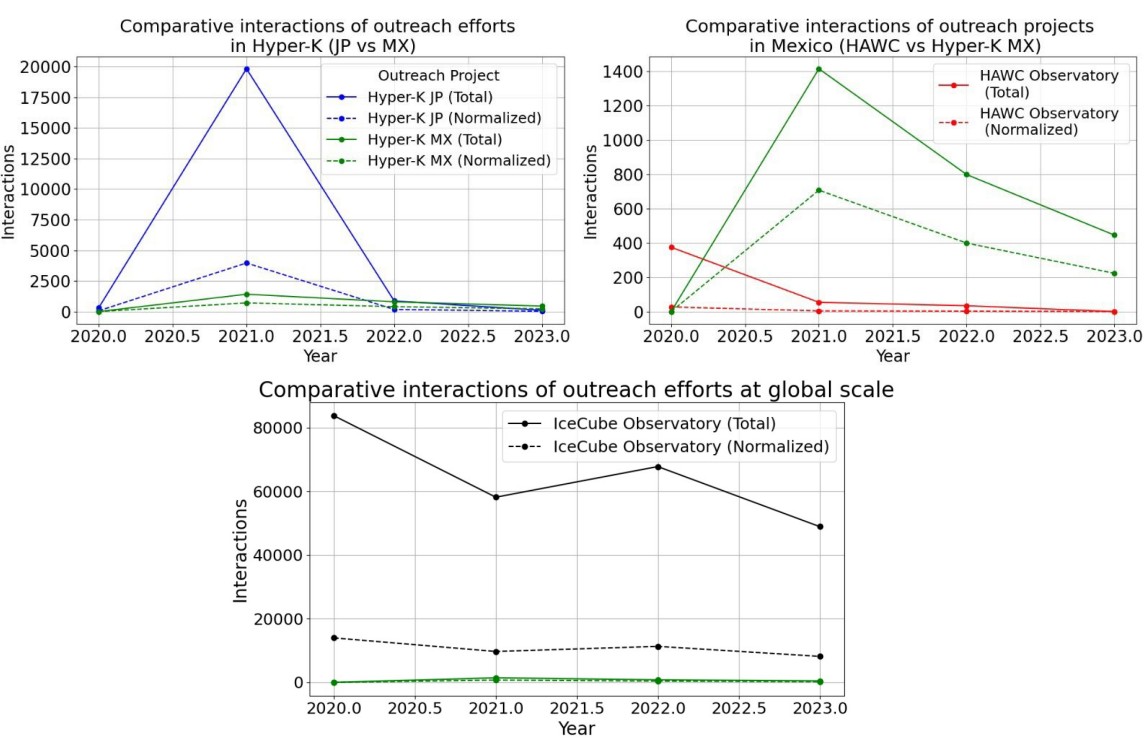

## 3.1 Mexico and Hispanic America Outreach

In 2024, despite the collaboration of Brazil in the Hyper-K project, a public search reveals that Outreach Hyper-K Mexico remains the only initiative of the Hyper-K project in the Americas. Therefore, to provide a comparative analysis, we need to look at another particle observatory within Mexico, such as the High-Altitude Water Cherenkov Observatory (HAWC)(6). The institutions managing the social media pages of the HAWC in Mexico [1] are led by the Universidad Nacional Autónoma de México (UNAM) and the Instituto Nacional de Astrofísica, Óptica y Electrónica (INAOE). At the top right of Fig.1 it is shown that Hyper-K MX generally achieves higher interactions than HAWC Mexico (HAWC MX).

## 3.2 Institutional Hyper-K Outreach

The main outreach initiative of the Hyper-K project (Hyper-K Japan) involves 5 main institutions: The University of Tokyo, the High Energy Accelerator Research Organization (KEK), the Japan Proton Accelerator Research Complex (J-PARC), the Kamioka Observatory, and the

---

[1]https://www.hawc-observatory.org/collaboration/

67 Institute for Cosmic Ray Research (ICRR). In the top left of Fig. 1, Hyper-K JP is shown to typ-
68 ically have more interactions than Hyper-K MX, except in 2022 for normalized interactions,
69 and in 2023, when Hyper-K MX exceeds in both total and normalized interactions.

## 3.3  Worldwide Outreach

71 To determine whether Outreach Hyper-K MX is having a significant global impact, we com-
72 pare its number of social media interactions with those of other neutrino experiments with an
73 international presence, such as the IceCube Neutrino Observatory, a detector located in Antarc-
74 tica and led by the University of Wisconsin-Madison [2] (USA)(7). At the bottom of Fig. 1, it
75 is shown that IceCube generally surpasses Hyper-K MX in both total and normalized annual
76 interactions.

## 3.4  Statistical analysis

78 Tab.1 shows unnormalized core participation metrics for key engagement measures, while
79 Tab.2 presents normalized per-participant counts to determine efficiency across outreach ini-
80 tiatives. Based on the EDA model, these metrics are essential for evaluating engagement suc-
81 cess and facilitating comparisons between initiatives (4). In Tab.3, the engagement rate is
82 provided for each type of post across different social media platforms. This rate reflects au-
83 dience interaction relative to the follower count within a specified timeframe, offering insight
84 into content appeal and reach on social media.

Table 1: Statistical Analysis of total social interactions (2020-2023).

| Observatory | mean | std | min | 25% | 50% | 75% | max | Total interactions |
|---|---|---|---|---|---|---|---|---|
| Hyper-K JP | 5276.25 | 9701.23 | 101.0 | 261.50 | 592.0 | 5606.75 | 19820.0 | 21105.0 |
| Hyper-K MX | 665.25 | 597.81 | 0.0 | 334.50 | 622.5 | 953.25 | 1416.0 | 2661.0 |
| HAWC | 115.75 | 174.264885 | 0.0 | 25.50 | 44.0 | 134.25 | 375.0 | 463.0 |
| IceCube | 64605.25 | 14904.84 | 48822.0 | 55801.50 | 62937.5 | 71741.25 | 83724.0 | 258421.0 |

Table 2: Statistical Analysis of normalized interactions (2020 to 2023).

| Observatory | mean | std | min | 25% | 50% | 75% | max | Normalized interactions |
|---|---|---|---|---|---|---|---|---|
| Hyper-K JP | 1055.25 | 1940.25 | 20.2 | 52.30 | 118.40 | 1121.35 | 3964.00 | 4221.00 |
| Hyper-K MX | 332.63 | 298.91 | 0.0 | 167.25 | 311.25 | 476.63 | 708.0 | 1330.5 |
| HAWC | 8.27 | 12.45 | 0.0 | 1.82 | 3.14 | 9.59 | 26.79 | 33.07 |
| IceCube | 10770.42 | 2480.08 | 8148.5 | 9303.13 | 10489.58 | 11956.88 | 13954.00 | 43081.67 |

85 Statistical analysis reveals differences in outreach interactions across projects. In Tab.1, the
86 IceCube Neutrino Observatory leads with a mean of 64,605.25 total interactions, highlighting
87 strong outreach effectiveness. By contrast, Hyper-K MX has a lower mean of 665.25, suggest-
88 ing room for improvement in engagement approaches. Tab.2 presents normalized interactions,
89 where the IceCube Neutrino Observatory maintains significant outreach with a normalized
90 mean of 10,770.42, while Hyper-K JP and Hyper-K MX show lower values, at 1,055.25 and
91 332.63, respectively. Tab.3 shows engagement rates in 2023 by platform, with Hyper-K MX
92 achieving high rates on YouTube (538.9%) and TikTok (340.7%), where audience interactions
93 are notably higher than on Facebook (7.2%).

---

[2]https://icecube.wisc.edu/collaboration/institutions/

Table 3: Engagement rate by each Outreach.

| Observatory | Facebook | Instagram | YouTube | X | TikTok |
|---|---|---|---|---|---|
| IceCube | 72.7% | 242.4% | 138.6% | 57.9% | - |
| HAWC | 0.0% | - | - | - | - |
| Hyper-K JP | 0.0% | - | 0.0% | 5.4% | - |
| Hyper-K MX | 7.2% | 80.9% | 538.9% | 8.5% | 340.7% |

In social media analysis, engagement rates can sometimes surpass 100% due to the way they are calculated, as we can see in Tab.3. This often involves measuring interactions as a percentage of impressions or reach instead of total followers. When users engage multiple times—by liking, commenting, or sharing on the same posts—the engagement rate can exceed 100%, particularly for content that goes viral or is viewed frequently. Research indicates that this elevated engagement typically signifies strong user interest and repeated exposure, rather than just unique interactions, offering a more nuanced understanding of online engagement trends (8).

Table 4: Outreach Hyper-Kamiokande Mexico's Facebook and Instagram accounts including top follower's information (Last 28 Days retrieved 25/oct/2024)

| | Top Locations (%) | | | | | | Age Range [1] | |
|---|---|---|---|---|---|---|---|---|
| | Mexico | USA | France | Germany | Colombia | Japan | Years | % |
| Instagram | 86.2 | 3.8 | 1.7 | 1.4 | 1 | - | 18-24 | 56.6 |
| Facebook | 309 | 12 | 0 | 3 | 0 | 3 | 25-34 | 42.8 |

[1] The age range percentages represent the largest follower demographics on each platform. For further information: http://hyperkamiokande.itesm.acsitefactory.com/es.

These results in Tab.4 evidence the disparity in outreach effectiveness and highlight the need for targeted strategies to enhance engagement, particularly for Hyper-K MX and HAWC. It is important to mention that our objectives focus on engaging an audience aged 18 and older, as our outreach aims to communicate the Hyper-K advances to a broader community.

# 4 Discussion

Across the nation, Hyper-K MX performs slightly higher social media engagement and online presence than the HAWC Observatory. In contrast to Hyper-K JP, Hyper-K MX has lower engagement overall, although it has outperformed Hyper-K JP in the past two years, less posting frequency is the possible reason behind these results. Globally, Outreach Hyper-K MX exhibits lower engagement and efficiency than the IceCube Observatory, as shown in Tab.2. From October 1 to October 25, 2024. Mexico has become the leading location for outreach engagement on Instagram and Facebook, comprising 86 2% of Instagram interactions and 309 interactions on Facebook. The United States and Germany represent 3.8% and 1.4% of Instagram interactions, along with 12 and 3 Facebook interactions, respectively. The 18-24 age group leads on Instagram with 56.6%, while the 25-34 age group is most active on Facebook with 42.8%. The gender distribution is similar on both platforms, with men representing 56.8% of interactions on Instagram and 56. 5% on Facebook and women representing 43.2% and 43.5%, respectively. It is important to note that the Hyper-K project is still under construction, which affects the frequency and scope of our content, while also highlighting future engagement opportunities.

Our efforts have elevated the national visibility of the project, engaged diverse audiences, and fostering scientific interest in Latin America. Using social media and community initiatives, Hyper-K Mexico is an impactful example of international scientific collaboration, showing growth in online engagement and setting a strong foundation for future outreach. However, Outreach Hyper-Kamiokande MX has experienced a low and declining trend in the last two years in annual interactions, exposing the need for revised engagement strategies to address this issue in the coming years. For instance, increasing the posting frequency, as seen with the outreach of IceCube Observatory, and adapting content to resonate with the Latin American culture could enhance engagement.

## 5   Conclusion

Hyper-K Mexico has made significant progress in increasing the project's national visibility and fostering public engagement through social media and community-focused initiatives. This success is particularly evident in its strong outreach engagement within Mexico, especially on Instagram and Facebook, where it has effectively connected with diverse demographics.

Although Hyper-K Mexico's global engagement remains lower than that of the IceCube Observatory, its recent performance has been notable, surpassing the annual interactions achieved by the outreach efforts of Hyper-K JP.

However, the declining trend in annual interactions over the past two years highlights the need for strategic adjustments. To address this, increasing posting frequency and developing content tailored to resonate with the Latin American audience will be essential. As Hyper-K continues its construction phase, these efforts will be crucial for building a solid foundation for future engagement and ensuring the project's long-term success in fostering scientific interest across Mexico.

## 6   Acknowledgements

MMDR acknowledges financial support from the Comission C4 of the International Union of Pure and Applied Physics (IUPAP) for attending ISVHECRI 2024. We also thank CONAHCYT for funding through the CF-2023-G-643 and CBF2023-2024-427 projects. Special thanks to ANDES Lab, fashion designer Panni Margot, the Comunidad de Aceleradores de Partículas del Tecnológico de Monterrey (CAPTEC), and the Sociedad de Alumnos de Ingeniería Física Industrial (SAIFI) for their support and collaboration in outreach events.

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
