# Peer review of "Outreach and Impact of the Hyper-Kamiokande experiment in Mexico and Latin America from 2020 to 2023 in social media."

_SciPost Physics Proceedings_

## Round 1 · Referee Report · Sei Kato (Referee 1) · 2024-12-27

Report

This article discusses the effect of outreach activities made by Hyper-K Mexico for public Latin America community during 2020 and 2023. The motivation of the article is clear and the results are interesting. In particular, it is clearly seen that the power of Mexico in outreach is very strong. I would recommend our editor to publish this article, after making some minor revisions and answering questions posted on "Requested changes."

Requested changes

  1. In abstract, authors should explicitly present the duration of interest for their research, from 2020 to 2023.
  2. L12: "... its broader implications. Through social media engagement..." -> "... its broader implications, through social media engagement..."
  3. For the sake of clarity, I recommend authors explicitly list all legends in the top right and bottom figures in Figure 1: authors should also add the legend "Hyper-K MX (Total)" and "Hyper-K MX (Normalized)" to the figures.
  4. It is interesting to see that the curve of IceCube is also in a descending trend during 2020 and 2023, despite their increasing (or constant) posting frequency, as written in L128-129. Do you have any idea why it is, and do you think you have something to learn from this tendency to improve the outreach by HK-MX?
  5. The meaning of "disparity" in L102 is not clear. Does it mean that Mexico far overwhelms Japan in terms of the number of interactions in Instagram and Facebook, while Japan has much higher number of total interactions than Mexico in 2021? I recommend authors add a simple explanation to clarify what "disparity" means.
  6. L112: "..., 2024. Mexico has become..." -> "..., 2024, Mexico has become…”
  7. L111-112: Authors mention that they focused on October 1 to October 25, 2024 to retrieve the data from Instagram and Facebook, which is not consistent with the caption of Table 4: "... (Last 28 Days retrieved 25/oct/2024)". I think authors should present the correct duration consistently in their article.

Recommendation

Ask for minor revision

  • validity: -
  • significance: -
  • originality: -
  • clarity: -
  • formatting: -
  • grammar: -

Author:  Judith Torres Jiménez  on 2025-01-17  [id 5133]

(in reply to Report 1 by Sei Kato on 2024-12-27)

Thank you so much for the comments.

Attachment:

Hyper_Kamiokande_Outreach_paper.pdf

Anonymous on 2025-01-21  [id 5138]

(in reply to Judith Torres Jiménez on 2025-01-17 [id 5133])

Der Torres Jiménez,
After reading the revised article, I confirmed that the revisions I proposed are properly accounted for. I will accept the article for publication only after correcting "Fig. ??" in L61, L67, and L75.

---

## Round 1 · Referee Report · Sei Kato (Referee 1) · 2025-1-21

Report

After checking the revised article, I confirmed that the revisions I suggested are properly accounted for in the article. I accept the article for publication only after correcting "Fig. ??" in L61, L67, & L75.

Recommendation

Publish (easily meets expectations and criteria for this Journal; among top 50%)

  • validity: -
  • significance: -
  • originality: -
  • clarity: -
  • formatting: -
  • grammar: -

Author:  Judith Torres Jiménez  on 2025-01-29  [id 5161]

(in reply to Report 2 by Sei Kato on 2025-01-21)

Thank you. Here is the updated version.

Attachment:

Hyper_Kamiokande_Outreach_paper__Copy_.pdf

---

## Editorial Decision

voting_in_preparation